# Mapping Grip Force Characteristics in the Measurement of Stress in Driving

**DOI:** 10.3390/ijerph20054005

**Published:** 2023-02-23

**Authors:** Yotam Sahar, Tomer Elbaum, Oren Musicant, Michael Wagner, Leon Altarac, Shraga Shoval

**Affiliations:** Department of Industrial Engineering & Management, Ariel University, Ariel 4076414, Israel

**Keywords:** grip force, stress, steering wheel, pedestrian, remote driving, driving simulation

## Abstract

Reducing drivers’ stress can potentially increase road safety. However, state-of-the-art physiological stress indices are intrusive and limited by long time lags. Grip force is an innovative index of stress that is transparent to the user and, according to our previous findings, requires a two- to five-second time window. The aim of this study was to map the various parameters affecting the relationship between grip force and stress during driving tasks. Two stressors were used: the driving mode and the distance from the vehicle to a crossing pedestrian. Thirty-nine participants performed a driving task during either remote driving or simulated driving. A pedestrian dummy crossed the road without warning at two distances. The grip force on the steering wheel and the skin conductance response were both measured. Various model parameters were explored, including time window parameters, calculation types, and steering wheel surfaces for the grip force measurements. The significant and most powerful models were identified. These findings may aid in the development of car safety systems that incorporate continuous measurements of stress.

## 1. Introduction

Grip force is an index that is based on the measurement of the pressure applied by a person’s hand on a given surface or handhold. Accumulated anecdotal evidence indicates the occurrence of ‘white finger syndrome’ among combat pilots, which describes their tendency to hold the aircraft’s steering handle very tightly during highly stressful situations, to the extent that blood is drained from the fingers, leaving the fingers pale. These reports laid the foundation for the research on grip force as a measure of stress [1].

Stress is a concept that has many definitions. Among these conceptualizations, there are two prominent models. The first is focused on the response to a threatening stimulus and is known as the ‘Fight-or-Flight’ response, which is the first stage of the general adaptation syndrome that regulates stress responses [2]. The second conception is cognitive and is referred to as the transactional model [3]. According to this model, stress occurs when there is a gap between one’s perception of his/her abilities and the perceived requirements of a given situation, provided that there are perceived consequences for that individual.

From the perspective of the central nervous system, stress is perceived and processed in various brain structures, including lower ones, such as the amygdala and the hypothalamus, and higher ones, such as the prefrontal cortex. The lower brain structures are characterized by a faster processing rate than the higher brain structures [4].

Stress during driving may cause a high mental workload [5] and lead to adverse effects [6] that, in turn, may decrease driver performance [7,8]. Therefore, the monitoring of drivers’ stress has the potential to improve drivers’ performance and safety through stress-targeted interventions, such as stress-adaptive car systems that modify the parameters of in-vehicle driver-aiding systems (DAS) [9]. A similar method was recently revealed to be efficient in improving psychomotor training through the use of adaptive automation capabilities combined with near-real-time stress measurement based on grip force [10].

Many physiological indices of stress have disadvantages, such as high measurement latency and intrusiveness [11]. However, grip force is an innovative index of stress that is unaffected by these drawbacks. Previous work on grip force as an index of stress was conducted by Wagner et al. [1] and Botzer et al. [12], who compared levels of grip force in tracking tasks with and without stress on the group as well as at the individual level. Additionally, Sahar et al. [11] measured grip force during driving and the performance of a braking task at various levels of intensity. The lessons learned from these studies indicate that grip force can be used to distinguish between states with and without stress [1], that a five-second time window of grip force data is sufficient for the detection of stress [12], and that stress can be detected through the steering wheel during driving using a two-second time window of grip force [11].

Nonetheless, these studies have several limitations. First, there are possible inertial effects (due to the braking events) on grip force that may occur instead of or in addition to the stress’s effect [11]. One way to avoid inertial effects on the driver’s grip force is to have the participant perform a driving task in a completely stationary environment (e.g., by driving a remotely controlled vehicle or a simulation). Another method used to isolate different effects on grip force is to increase the resolution of its measurement, so that the grip force that is closely correlated with an irrelevant factor, such as inertia during a car braking event, can be disregarded.

A second limitation is that the manipulations might induce other states rather than stress, such as a mere workload state. One example of a driving-related stressor is a pedestrian crossing the road quickly and without warning at various distances from the vehicle. Such a manipulation was previously found to cause stress in driving simulator settings [13,14]. Furthermore, compared to simulator driving, the performance of the same driving task in a real vehicle was found to elicit stress [15]. Thus, comparing the same driving task in both driving modes (simulated or on-road) may be used as another two-level stress manipulation. However, this stressor remains throughout the driving task, unlike the crossing pedestrian manipulation, which occurs for a shorter period during the driving task.

Another limitation lies in the fact that only a limited variety of mathematical operations and statistical methods have been explored regarding grip force. The quantification of this relatively new measure may include more efficient preprocessing and statistical methods, depending on the characteristics of the stressor and the task at hand. From similar studies that explored physiological measures of stress, it appears that the main parameters used for characterizing the state of stress according to raw data over time are the size of the time window (hereafter, ‘Time window’), the offset of the time window from the stressful event (hereafter, ‘Offset’), and the mathematical operation performed on the raw data (hereafter, ‘Calculation’) [16]. Specifically, in the measurement of stress according to grip force data, the handhold surface that the grip force acts upon is affected by stress (hereafter, ‘Surface’) is also an important characteristic of this index.

The current research aimed to address the abovementioned limitations by isolating the different effects on grip force using completely stationary driving tasks; that is, driving in a simulator and controlling a remote vehicle. In addition, by utilizing sizable stimuli, another aim of this study was to explore various parameters in the calculation of grip force. This will enable the formulation of models describing the relationship between grip force and stress. Our hypothesis is that grip force, measured through the steering wheel, can be used to distinguish between two levels of stress during driving. Specifically, we hypothesized that:

**H1.** 
*Grip force calculations for a pedestrian crossing event will be higher for a closer crossing distance compared with a further crossing distance of the pedestrian.*


**H2.** 
*Grip force calculations for a pedestrian crossing event will be higher in a remote driving mode compared with a simulated driving mode.*


## 2. Materials and Methods

### 2.1. Participants

Forty-two participants took part in this study, of whom thirty-nine completed the entire experimental procedure, with full data received. The participants were bachelor students at Ariel University, including 26 males and 13 females between the ages of 23 and 47 (average 26.87, SD 3.90), and had had a private car driver’s license for at least three years (average 8.55, SD 4.38).

### 2.2. Apparatus

#### 2.2.1. Modes of Driving

Two modes of operation were applied in the current experiment, as illustrated in Figure 1. One was the control of an instrumented Kia Nero (hereafter, the ‘Mobile-Lab’, see Figure 2) from a distance (hereafter, ‘Remote driving’), and the other was the operation of a virtual car in a simulated driving environment (hereafter, ‘Simulated driving’). The operation station (for both remote driving and simulated driving) was located in a laboratory room. It consisted of a SPEEDSeat SS-V1 gaming seat, three G27FC gaming monitors, a Logitech G29 gaming steering wheel and pedals, and a PC with an Intel^®^ Core™ i7-10700K processor and GeForce RTX 3080 graphics card. The simulation environment was supplied by Cognata Studio.

The experiment was conducted using a remotely controlled vehicle, the Mobile-Lab [11], which was equipped with sensors for monitoring the vehicle and road environment, including inertial measurement units (IMU), lidar, and GPS antennas. Three cameras positioned on the mid-front of the car roof rack transferred live video to the remote driving station. A remote driving system (supplied by the ‘Drive-U’ company) was installed in the Mobile-Lab, controlling its steering wheel and pedals. The remote driving system’s average latency was 200 milliseconds. The use of remote driving instead of a naturalistic driving mode allowed us to inspect the effects of the actual consequences of the driver’s actions, as opposed to simulative driving, without the interference of other factors that exist in naturalistic driving, such as accelerations, acting on the driver’s body.

#### 2.2.2. Grip Force Measurement System

A self-developed measurement system (see Figure 3) was used to measure the grip force on the steering wheel. This system consisted of sixty force-sensitive resistor (FSR) sensors, which were divided between three steering wheel surfaces (close to the driver, on the circumference of the steering wheel, and on the surface of the steering wheel, that is, far from the driver). These FSR sensors were sampled using an ESP32 board with a 20 Hz sample rate, and the data were transmitted in real time via Wi-Fi to the main experimental computer, with a transmission latency of one millisecond.

#### 2.2.3. Stress Manipulations

##### Pedestrian Crossing Distance

One stress manipulation was the appearance of a pedestrian dummy crossing the road quickly and without warning at two distances from the front of the vehicle (see Figure 4). A similar manipulation was previously used in driving studies, such as that conducted by Daviaux et al. [17], who explored a driving task in a simulator while the pedestrian crossed the lane at one of two distances from the driver’s car: fifty meters from the car or right in front of it. The skin conductance response (SCR) was used to measure the driver’s stress level, which was significantly higher under the closer crossing condition.

In the current study, in the remote driving condition mode, a pedestrian dummy (fixed to a skateboard that was manipulated by cables) was moved into the driving route at one of two distances in front of the vehicle: four meters (hereafter, ‘close’) or eight meters (hereafter, ‘far’). The ‘close’ crossing condition distance was determined according to the vehicle’s maximal speed (at which the participants were instructed to drive, on which basis the vehicle’s gear was set) of 20 km/h in order to force the driver to perform emergency braking [18]. The ‘far’ crossing condition distance was set to eight meters, a closer distance than those used in previous studies [17], to inspect the ability of the grip force index to distinguish between closer levels of stress; that is, to determine the resolution of the index. These two distances defined the two stress levels that we hypothesized, namely, higher stress under the ‘close’ condition and lower stress under the ‘far’ condition. In addition to the manipulated pedestrian dummy, stationary pedestrian dummies were placed as distractions along the driving route to prevent the participant from predicting when the crossing event would occur. Under the simulated driving condition, equivalent conditions (distances and distractions) were defined.

The pedestrian crossing time was determined as a temporal reference point for the retrieval of the grip force data. The exact crossing start time was extracted manually from the screen capture and recorded on the same computer as the physiological indices.

##### Driving Mode

The second stress manipulation was the driving mode, which was either remote driving or simulated driving. Since the consequences of a possible accident involving a crossing pedestrian in remote driving are more tangible compared with the same event in simulated driving, stress in the former driving mode is hypothesized to be greater than that in the latter. In both driving modes, the operation means (i.e., the steering wheel and pedals), the viewing screens, and the seat were the same, as the participant remained seated in the same operation station, and only the driving mode was altered.

### 2.3. Procedure

The participants, upon arriving individually at the laboratory, received a brief on the experimental procedure, filled out an informed consent form, and completed a five-minute simulated driving training phase in an urban setting. Afterwards, the SCR electrodes were attached (SCR measurements were used for validation purposes; see Appendix A), and the participant completed four experimental trials, all consisting of driving through a slalom route between cones (see Figure 5). At the end of each route, a pedestrian crossing manipulation occurred. All participants completed two simulated driving and two remote driving trials, where both crossing distances appeared in each driving mode. The participants were randomly assigned to different orders of the four experimental conditions (2 driving modes × 2 crossing distances).

### 2.4. Data Collection and Calculation Parameters

To explore diverse means of quantifying the raw grip force measurements, multiple parameters were applied, as illustrated in the infographics in Figure 6. The parameters were the time window, offset from the crossing event, the surface of the steering wheel, and the combination of calculations (for each sample and for all samples in the entire time window).

Grip force raw data (see in Figure 7) were sampled at 20 Hz. For each crossing event, the initial sample (t_1_) was defined as 26 different offsets (0 to 5 s at 0.2 s intervals) from the onset of the event. For the zero offset, there were 25 different time windows (0.2 to 5 s at 0.2 s intervals). For offsets greater than zero, there were 5 different time windows (1, 2, 3, 4, or 5 s). For each grip force sample (either for data from all 60 FSR sensors or for each of the 3 surfaces of the steering wheel), a calculation was applied (Calc_a_: maximum, mean, or standard deviation). A second calculation was applied to the outcome of Calc_a_ of all the samples in the time window (Calc_b_: maximum, mean, standard deviation, or median).

This summarization process was performed with all possible permutations for each crossing event and for each participant. The data consisted of a total of 156 crossing events (39 participants who completed the task) × 2 (driving modes) × 2 (crossing distances)). A linear mixed model (LMM) was selected for the analysis (see also [1,10,11,12]), with the participant as the random effect to account for interpersonal differences in grip force, according to the formula in Equation (1):(1)$$Grip\;Force=\beta_{0}+\beta_{1}*Stressor+Participant_{i}+\varepsilon$$
where ‘*Grip Force*’ represents the combination of all parameters of the calculation of the grip force (i.e., the calculation for each sampling event, calculation for all samples in the time window, time window width, and time window offset), ‘*Stressor’* represents the level of one of the stressors (i.e., the pedestrian cross distance or driving mode), and ‘*Participant*’ represents the random effect of the participant, with ‘*i*’ representing the specific participant.

## 3. Results

In order to examine the various combinations of calculations and time window parameters, multiple LMM analyses were conducted in accordance with the formula in Equation (1). For zero offset of the time window from the crossing even, and for the ‘Crossing distance’ stressor, 300 combinations were explored: 4 (window calculations) × 3 (single sampling of all 60 sensors) × 25 (time windows). Of these analyses, 53 were significant (*p* < 0.05), as can be seen in Figure 8. The smallest time window width that was significant was 2 s, based on a window ‘Mean’ calculation and sampling ‘Maximum’ calculation. A significant main effect of the ‘close’ crossing distance condition on the grip force was identified, t_(108)_ = −2.021, *p* = 0.046, with an explained variation of R^2^ = 0.276 and effect size of Cohen’s D = 0.387. A pattern can be determined from the graph in Figure 8, indicating that the analyses of the time window width, ranging from 2 to 5 s, were significant, with peak significance for the 3 s time window width. It should be noted that larger time window widths may have yielded significant models. However, this research was focused on a range of widths between zero and five seconds, based on previous work [11,12].

For the zero offset, the most significant combination of parameters was the 3 s-width time window, based on a window ‘Mean’ calculation and sampling ‘Maximum’ calculation. A significant main effect of the ‘close’ crossing distance condition on the grip force was identified, t_(108)_ = −2.364, *p* = 0.019, with an explained variation of R^2^ = 0.368 and effect size of Cohen’s D = 0.453, which indicates that the close crossing distance caused greater grip force, as can be seen in Figure 9.

From the investigation of all offsets of the time window and all possible combinations of all the other parameters, as mentioned earlier (i.e., all calculation types, time window widths, and stressors), it was found that various parameter combinations yielded significant results. Table 1 includes a summary of all the significant models according to their parameters. These models in Table 1 are characterized by a higher statistical strength compared to the others. For the stressor ‘Crossing distance’, the strongest of these models is that based on the ‘Mean’ of the window and ‘Maximum’ of the sampling, with a 2 s-width time window and an offset of between 1.8 and 4.2 s, which has an explained variation of R^2^ between 0.41 and 0.82 and an effect size of Cohen’s D between 1.041 and 1.509. For the stressor ‘Driving mode’, there are three combinations of calculations that are significant, including the ‘Median’ of the window with the ‘Maximum’ of the sampling, the ‘Median’ of the window with the ‘Standard deviation’ of the sampling, and the ‘Median’ of the window with the ‘Mean’ of the sampling. The widths of the time window are between 4.2 and 5 s for all three combinations, all with zero offset. The explained variation for these models is indicated by an R^2^ between 0.24 and 0.27 and an effect size of Cohen’s D between 0.408 and 0.467.

The aggregation of the parameters of these models indicates that, for the ‘Crossing distance’ stressor, there are various relevant time window widths, with the statistically strongest effects observed for the smaller widths. On the other hand, for the ‘Driving mode’ stressor, most of the time window widths fall at the higher end of the inspected range (i.e., 4.2–5 s).

The inspection of grip force on the different surfaces of the steering wheel included, in addition to this parameter, its combination with all the previous ones (i.e., all calculation types, time window widths, and offsets, as well as stressors). Of the possible combinations of parameters, significant models were observed for the ‘Driving mode’ alone (i.e., ‘Remote driving’ versus ‘Simulated driving’), with no effects of the time window width and with zero offset alone. Table 2 includes the parameters of these models. Observing these results, different directions of the stressor effect on the grip force combination of calculations interacting with the surface of the steering wheel are apparent. For the far surface of the steering wheel, the grip force is increased in the ‘Remote driving’ mode compared with the ‘simulated driving’ mode, while for the surface of the steering wheel that is near to the driver, it is lower. The explained variation for these models is indicated by an R^2^ between 0.52 and 0.60 and an effect size of Cohen’s D between 0.426 and 0.470.

Each participant performed the driving task (in both modes) four times, in which the incident of the pedestrian crossing the lane occurred each time. In order to examine the effect of the first occurrence of the crossing event itself on the grip force, LMM analyses were conducted for all the grip force calculation combinations, as previously described, as well as all the time window parameters (widths and offsets), as a factor of the crossing event order (the first against the last of the three crossing events), in accordance with the formula in Equation (2). All analyses were non-significant.
(2)$$Grip\;Force=\beta_{0}+\beta_{1}*Crossing\;Order+Participant_{i}+\varepsilon$$

To inspect possible effects of participants’ background characteristics (i.e., gender, age, and years of driving experience) on the grip force, LMM analyses were conducted with the participant as the random effect, including interactions between these parameters. None of these analyses were significant. Additionally, all the interactions were examined (i.e., all combinations of the crossing distance, driving mode, crossing order, and background characteristics); however, none of these analyses were significant.

## 4. Discussion

We explored multiple models to describe the relationship between grip force and stressful events or states in order to locate the mathematical and statistical models most suitable for the measurement of stress according to the grip force. Our findings (see Table 1 and Table 2) showed that there are several significant models, some of which have a high statistical power, compared with other physiological indices of stress, such as cortisol levels [19] and heart rate measures [20]. These findings suggest that grip force on the steering wheel is a suitable measurement method for use as a stress index for driving tasks, in accordance with our previous findings [11]. Furthermore, since the two distances used for the ‘Crossing distance’ stressor were relatively close to one another [17], it can be concluded that grip force, as a stress index, is suitable for distinguishing between relatively close levels of stress.

Of the two stressor manipulations that were used, the ‘Crossing distance’ stressor was found to be significant, with a higher statistical power in the narrower time windows, while the ‘Driving mode’ stressor was significant in the case of the wider time windows (see Table 1). These two stressors can be distinguished in terms of their duration. That is, the ‘Driving mode’ stressor was characterized by a relatively long duration in this experiment; overall, the situation lasted for a number of minutes. On the other hand, the ‘Crossing distance’ stressor was characterized by a noticeably short duration, i.e., a few seconds. These different time periods of stressor events correspond with the time window widths observed in the models, a fact that strengthens these findings. In addition, the definitions of stress are also reflected in these time period differences. The transactional model of stress involves cognitive processes [3], which are carried out in the prefrontal cortex [21]. On the other hand, the ‘Fight-or-Flight’ stress response takes place in the sub-cortical brain structures (i.e., the amygdala and the hypothalamus), which are characterized by a faster processing speed than the cortical structures [21]. Therefore, it can be argued that short-term stressors, such as the ‘Crossing distance’, invoke the ‘Fight-or-Flight’ response, which, in turn, is reflected in certain grip force models, while states that induce long-lasting stress, such as the ‘Driving mode’ stressor, initiate the transactional model assessment processes, which, in turn, are reflected in other distinct grip force models.

Another finding that can aid in the appropriate choice of parameters in the study of grip force through stress measurements in driving scenarios is reflected in Table 2. According to these findings, the steering wheel surfaces reflect a reverse grip force reaction according to the given model. Models with the ‘Mean’ calculation of the time window showed an increase in the grip force in the stressful condition, while models with the ‘Maximum’ calculation of the time window showed a decrease under this condition. It should be noted that these models were independent of the width of the time window. This lack of effect of the time window width may be explained by the lasting effect of the relevant stressor (i.e., ‘Driving mode’), which remained the same throughout the driving task. In addition, since the various steering wheel surfaces were not significant in influencing the models of the effect of the ‘Crossing distance’ on grip force, a possible conclusion is that short-term stressors may not be expressed in the grip force using specific steering wheel surfaces.

The lack of evidence for the effect of the order of the crossing event on the grip force, although we cannot rule out this possibility entirely, nevertheless suggests the weakness of this effect. The argument that the first occurrence of the crossing event evoked higher levels of stress than the latter three crossing events, beyond the two levels of this manipulation (i.e., the ‘Close’ and the ‘Far’ conditions), is not supported. Thus, at the very least, the findings regarding the ‘Crossing distance’ manipulation effect on the grip force are neither contradicted nor refuted.

Our findings were independent of the possible effects of interpersonal background characteristics, such as age, gender, and driving experience. Therefore, potential alternative explanations of our results in such terms are not supported. Additionally, none of the possible interactions between these background parameters and the model parameters (i.e., the window and calculation options) were significant. Thus, the effects that we identified were independent of mutual influence, i.e., the effect of the ‘Driving mode’ on the grip force was beyond the distinct levels of the ‘Crossing distance’ stressor, and vice versa.

There are several limitations to the current study. First, the characteristics of the participants in the sample were relatively homogenous; that is, all the participants were students at the same institute. It is possible that other populations would yield different patterns. Future studies may benefit from examining the models explore here with other populations or, alternatively, with more heterogenic samples.

A second restraint was the reuse of a single dataset in the current study. The purpose of this research was the exploration of various mathematical and statistical models of the relationship between levels of stressors and grip force. In order to enable the comparison of the assorted models with multiple combinations of parameters, the same data were employed. However, this reuse of data may have skewed the results according to specific attributes of the current data. Researchers are encouraged to examine the validity of these models with different datasets. Should the current results be replicated, their validity would be reconfirmed.

A third limitation concerns the accuracy and validity of the results of this study. Since the explained variable in our models was grip force, a physiological index of stress, its accuracy should be verified according to a gold standard used in this field, such as cortisol or electroencephalogram (EEG) indices. Although grip force was validated here according to EDA, researchers are encouraged to use other indices in future studies so as to further examine this issue.

A fourth limitation stems from the physical setting of the operation station used here. In order to isolate the measurement of grip force and avoid external influences such as the effect of acceleration on the driver’s body, which, in turn, would be expressed in the grip force, we used a stationary operation position for the driver in both the simulated driving mode and remote driving. Nonetheless, the results reported here are restricted to immobile environments and should not be generalized to mobile settings, such as a driver in a moving vehicle or naturalistic driving. Future studies could verify these results by reproducing this experiment in mobile settings.

## 5. Conclusions

The aim of this research was to map characteristics that are relevant to the relationship between stress and grip force in driving tasks. The results indicate that there are various possible models that can be used to describe this relationship, some of which are more statistically powerful than others. The most relevant parameters of these models are those related to the time window (i.e., offset and width) and the calculations (of each sampling of the sensors and of the entire time window). Additionally, the time duration of the stressful event should be considered so that the appropriate model parameters can be applied. Specifically, for persisting stressful driving situations, the surface of the steering wheel on which the grip force is measured should be selected carefully, combined with the proper calculation method for the grip force, according to the recommendations based on our results.

The broad range of possibilities for the modeling of the relationship between grip force and stress in driving tasks demonstrated here will allow researchers and professionals to use grip force for the purpose of measuring stress in a non-intrusive manner and with relatively short latencies.

## Figures and Tables

**Figure 1 ijerph-20-04005-f001:**
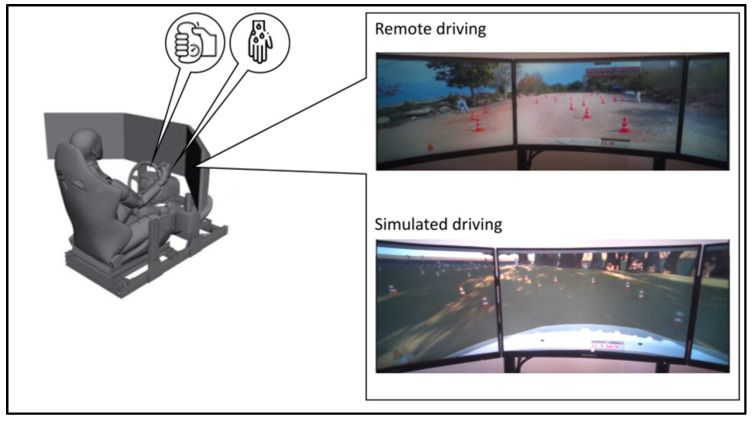
Illustration of the operator station used to operate both the remotely controlled vehicle and the driving simulation (with the same HMI) while grip force on the steering wheel and electrodermal activity are measured.

**Figure 2 ijerph-20-04005-f002:**
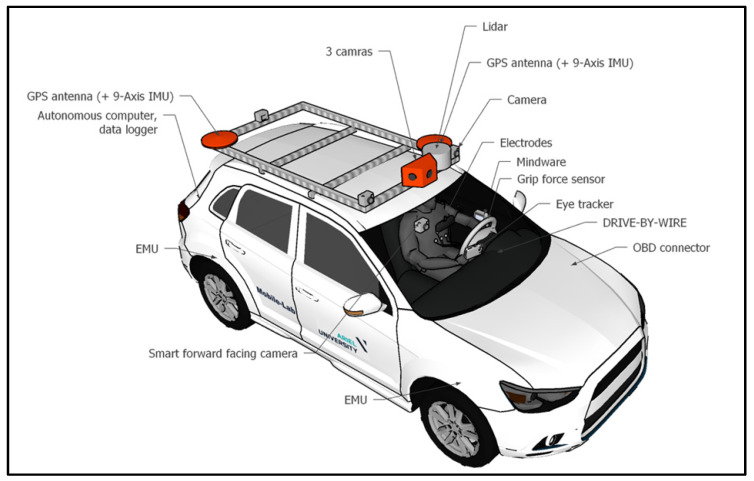
Ariel University’s Mobile-Lab, used in the current research as the remotely controlled vehicle, is equipped with a three-camera set transmitting the live video to the remote driving station, as well as a remote driving system.

**Figure 3 ijerph-20-04005-f003:**
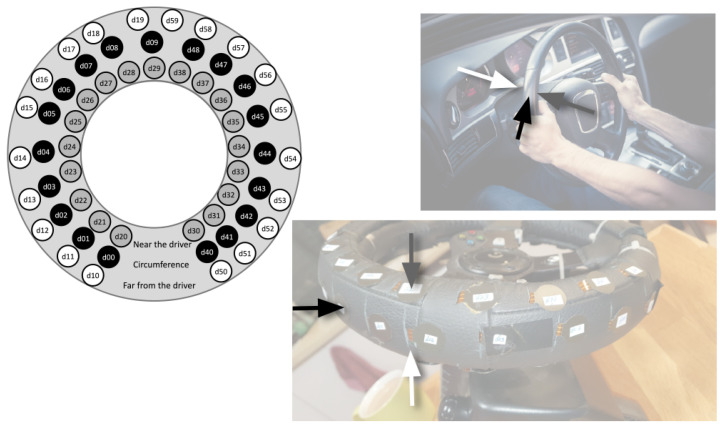
The steering wheel was embedded with 60 FSR sensors, which were grouped on three surfaces: close to the driver (grey circles in the illustration), on the circumference (black circles in the illustration), and on the surface far from the driver (white circles in the illustration).

**Figure 4 ijerph-20-04005-f004:**
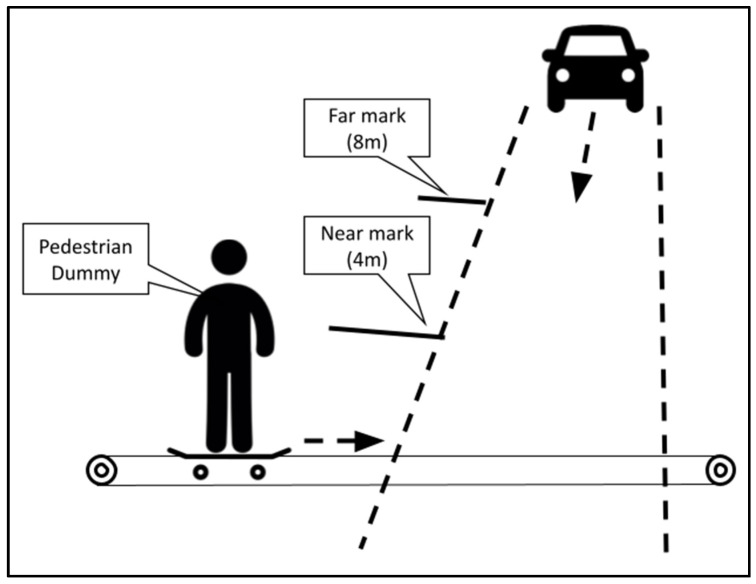
Pedestrian dummy crossing manipulation, consisting of a pedestrian dummy (fixed on a skateboard moved by cables) being moved into the driving route at one of two distances in front of the vehicle: 4 m (close) or 8 m (far).

**Figure 5 ijerph-20-04005-f005:**
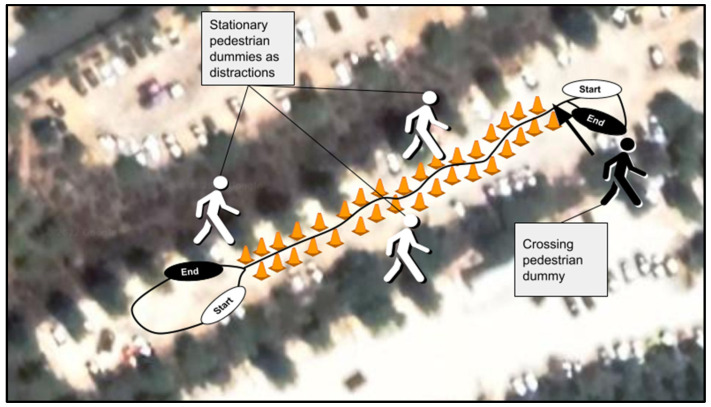
Driving route (for both remote driving and simulated driving conditions), consisting of road cones marking a slalom route with stationary pedestrian dummies as distractions, a U-turn marking and a crossing pedestrian dummy stress manipulation.

**Figure 6 ijerph-20-04005-f006:**
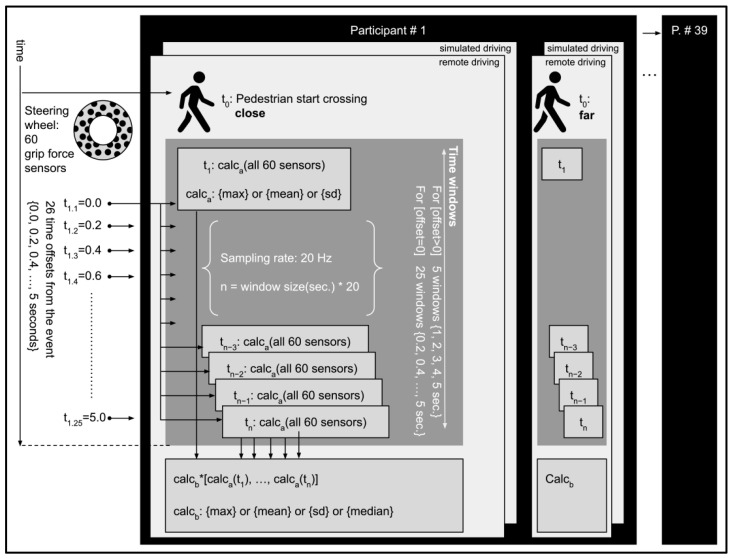
Grip force data collection and calculations. As illustrated, for each participant and each pedestrian crossing event, grip force data were calculated (Calc_a_: maximum, mean, or standard deviation) for all sensors or for each surface at each sampling instance. A second calculation was applied to all samples in the time window (Calc_b_: maximum, mean, standard deviation, or median).

**Figure 7 ijerph-20-04005-f007:**
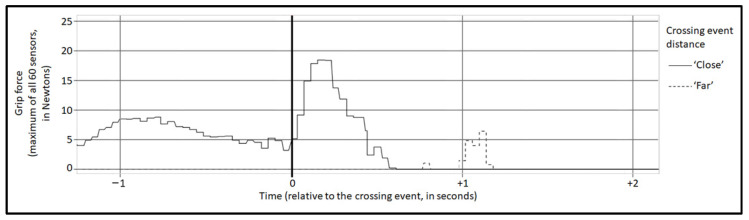
Raw grip force data from two crossing events of a single participant (randomly selected). *X*-axis represents time in relation to the crossing event (minus = before the event). *Y*-axis represents the maximal grip force at each measurement instance for all 60 sensors. The continuous line represents a ‘Close’ crossing event, while the dashed line represents a ‘Far’ crossing event.

**Figure 8 ijerph-20-04005-f008:**
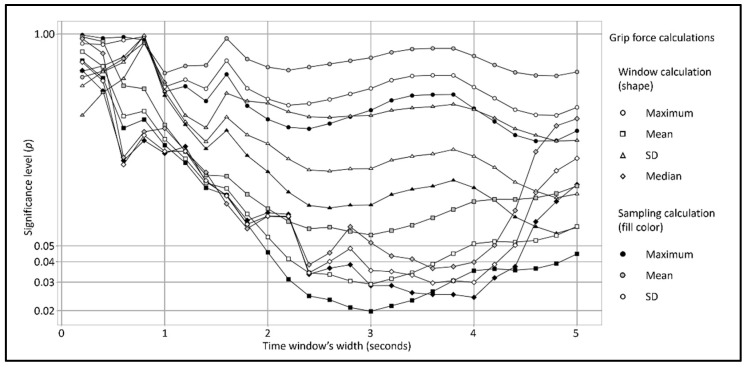
Significance level (*p*) of LMM analyses of grip force as a function of the crossing distance with the participant as the random effect, various time window widths, and no offset from the crossing event. *X*-axis expresses the width (0.2 to 5 s, 0.2 s increments), and *Y*-axis expresses the significance level (*p*), logarithmically scaled for representation. The shape type represents the window calculation (circle = maximum, square = mean, triangle = SD, rhombus = median), and the fill color represents the sampling calculation (black = maximum, grey = mean, white = SD).

**Figure 9 ijerph-20-04005-f009:**
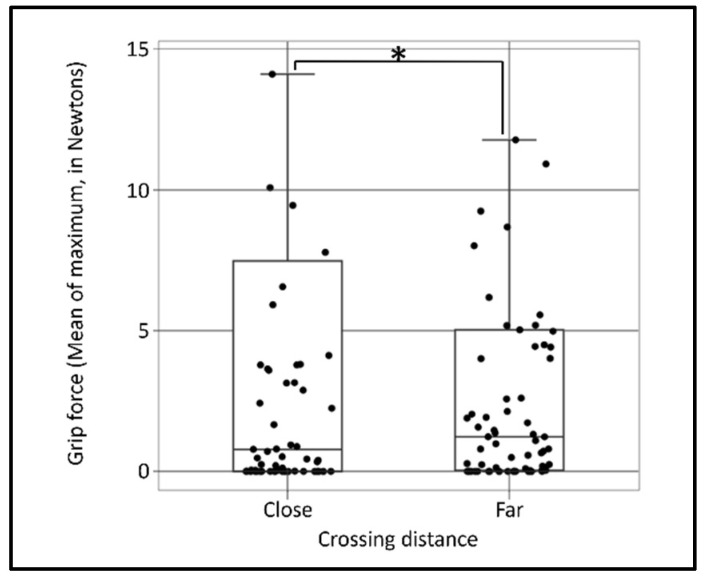
Grip force (window calculation: mean, single measurement event calculation for all 60 sensors; maximum in Newtons) as a function of the crossing distance. Boxes represent the inter-quartile range (IQR = Q1 to Q3) of the group, middle horizontal line represents the group’s median, upper line represents the largest value that is less than the upper quartile plus 1.5 times IQR, and lower line represents the smallest value that is greater than the lower quartile minus 1.5 times IQR. Asterisk (*) denotes a level of significance of *p* < 0.05.

**Table 1 ijerph-20-04005-t001:** Grip force calculations for all surfaces combined (60 sensors), as affected by one of two stressors. LMM model *: $Grip\;Force\;calculation=\beta_{0}+\beta_{1}*Stressor+Participant_{i}+\varepsilon$.

Calculation **	Stressor		Reaction to Stressor	Time Window Width (Seconds)	Time Window Offset (Seconds)	Conditional R2	Cohen’s D
	Crossing Distance	Driving Mode					
Mean of max	✓		↑	2–5	0	0.28–0.37	0.387–0.453
Max of mean	✓		↑	1	4.2	0.44	1.071
				2	1.8–4.2	0.41–0.82	1.041–1.509
				3	1–2.8	0.53–0.87	1.173–1.540
				4	0.2–2.6	0.53–0.73	1.206–1.388
				5	1.2–1.6	0.56–0.60	1.223–1.317
Mean of SD	✓		↑	2.2–3.8	0	0.32–0.38	0.389–0.424
Max of SD	✓		↑	2	2–2.6	0.69–0.71	1.350–1.355
				3	1.2–2.6	0.58–0.79	1.324–1.459
				4	0.2–2.4	0.54–0.79	1.233–1.353
Max of max	✓		↑	2	2.6	0.69	1.344
				3	1.2–2.6	0.58–0.79	1.312–1.453
				4	0.2–2.4	0.54–0.78	1.306–1.347
Median of max	✓		↑	2.4–4.4	0	0.21–0.30	0.401–0.437
				2	0.6	0.58	1.197
		✓	↑	4.2–5	0	0.24–0.26	0.408–0.465
Median of SD	✓		↑	2.4–4.2	0	0.20–0.30	0.382–0.421
				2	0.6	0.58	1.212
		✓	↑	4.2–5	0	0.25–0.26	0.417–0.465
Median of mean	✓		↑	2.4–4	0	0.18–0.29	0.386–0.405
				1	1.2	0.50	1.123
				2	0.6	0.62	1.295
		✓	↑	4.2–5	0	0.24–0.27	0.431–0.467
SD of mean	✓		↑	1	2	0.41	0.863
				4	0.2–2	0.21–0.67	0.854–1.253
				5	0.2–1.6	0.32–0.52	0.897–1.066

* The table includes significant models only (*p* < 0.05). ** Calculation format: <Calculation on entire window> of <Calculation on a specific point in time>.

**Table 2 ijerph-20-04005-t002:** Grip force calculations for each steering wheel surface, as affected by the stressor ‘Driving mode’ alone. LMM model *: $Grip\;Force\;calculation=\beta_{0}+\beta_{1}*Driving\;mode+Participant_{i}+\varepsilon$.

Index		Reaction to Stressor	Time Window Width (Seconds)	Time Window Offset (Seconds)	Conditional R2	Cohen’s D
Calculation **	SW *** Surface					
Mean of max	Far from the driver	↑	2–5	0	0.60	0.470
Mean of mean	Far from the driver	↑	2–5	0	0.55	0.426
Mean of SD	Far from the driver	↑	2–5	0	0.60	0.464
Max of max	Near the driver	↓	2–5	0	0.50	0.428
Max of mean	Near the driver	↓	2–5	0	0.53	0.449
Max of SD	Near the driver	↓	2–5	0	0.52	0.437

* The table includes significant models only (*p* < 0.05). ** Calculation format: <Calculation on entire window> of <Calculation on a specific point in time>. *** SW = steering wheel.

## Data Availability

Raw data were generated at Ariel University. The data of this study are available from the corresponding author upon reasonable request, considering the privacy of the research participants.

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
