# Peer review of "Mapping Grip Force Characteristics in the Measurement of Stress in Driving"

_ijerph, 2023, doi:10.3390/ijerph20054005_

Round 1
Reviewer 1 Report
This paper investigated the grip force characteristics in the measurement of stress in driving. The topic is meaningful. But I think some aspects should be revised.
1. The participants are all bachelor students at Ariel University. I think some skilled drivers should be included in the experiment.
2. There are two modes of driving, one is controlling an instrumented car from a distance, another is simulated driving. I’m confused that why don’t you use naturalistic driving instead of remote driving. Since remote driving is another kind of simulated driving, it cannot represent the real driver behavior in naturalistic driving.
3. Page 11, line 283, why do the possible combination of parameters only for the “Driving mode”? Can the simulated mode also calculated as a comparison?
Reviewer 2 Report
In this paper, the authors attempt to map the various parameters affecting the relationship between grip force and stress in driving tasks. This research is very interesting and has potential for publication at IJERPH. However, my following concerns need further clarification.
1. In Section 1, it is necessary to further clarify the relationship between the hypothesis or research scheme proposed in this study and the limitations of the current relevant research. The current statement in this paper cannot explain the direct contribution of this study to the solution of the shortcomings of other studies.
2. Please explain why the experiment was conducted based on the remote driving platform. Since remote driving is rarely used in daily driving activities, it is more often used in special scenarios, such as vehicle operation scenes in mining areas or mines. The application significance of remote driving in common traffic scenarios is not obvious.
3. As for the question about the distance between pedestrians and vehicles. Why is the distance between pedestrians and experimental vehicles set to 4 meters and 8 meters, and what is the standard for selecting the distance? Because the difference between 4 meters and 8 meters is not obvious.
4. Finally, the authors should further explain how to verify the accuracy and reliability of the research results.
Round 2
Reviewer 1 Report
The authors have revised the paper properly. I think the article can be accepted now.
Reviewer 2 Report
Thank the authors for their replies to the review comments. I think the manuscript has beenimproved and can be considered for publication.